# EMPHASIZING DISCRIMINATIVE FEATURES FOR DATASET DISTILLATION IN COMPLEX SCENARIOS

## ABSTRACT

Dataset distillation has demonstrated strong performance on simple datasets like CIFAR, MNIST, and TinyImageNet but struggles to achieve similar results in more complex scenarios. In this paper, we propose **EDF** (**e**mphasizes the **d**iscriminative **f**eatures), a dataset distillation method that enhances key discriminative regions in synthetic images using Grad-CAM activation maps. Our approach is inspired by a key observation: in simple datasets, high-activation areas typically occupy most of the image, whereas in complex scenarios, the size of these areas is much smaller. Unlike previous methods that treat all pixels equally when synthesizing images, EDF uses Grad-CAM activation maps to enhance high-activation areas. From a supervision perspective, we downplay supervision signals that have lower losses, as they contain common patterns. Additionally, to help the DD community better explore complex scenarios, we build the Complex Dataset Distillation (Comp-DD) benchmark by meticulously selecting sixteen subsets, eight easy and eight hard, from ImageNet-1K. In particular, EDF consistently outperforms SOTA results in complex scenarios, such as ImageNet-1K subsets. Hopefully, more researchers will be inspired and encouraged to improve the practicality and efficacy of DD. Our code and benchmark will be made public.

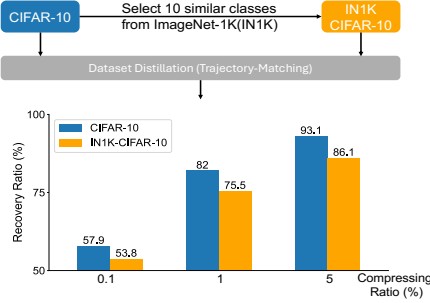

(a) The performance of dataset distillation drops remarkably in complex scenarios.

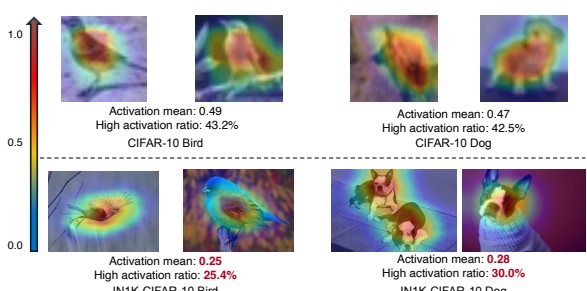

(b) Images from IN1K-CIFAR-10 have much lower activation means and smaller highly activated areas.

Figure 1: **(a)** DD recovery ratio (distilled data accuracy over full data accuracy) comparison between CIFAR-10 and IN1K-CIFAR-10. We use trajectory matching for demonstration. **(b)** Comparison between Grad-CAM activation map statistics of CIFAR-10 and IN1K-CIFAR-10. The ratio refers to the percentage of pixels whose activation values are higher than 0.5.

## 1 INTRODUCTION

Dataset Distillation (DD) has been making remarkable progress since it was first proposed by Wang et al. (2020). Currently, the mainstream of DD is matching-based methods (Zhao et al., 2021; Zhao & Bilen, 2021b;a; Cazenavette et al., 2022), which first extract patterns from the real dataset, then define different types of supervision to inject extracted patterns into the synthetic data. On several simple benchmarks, such as CIFAR (Krizhevsky, 2009) and TinyImageNet (Le & Yang, 2015), existing matching-based DD methods can achieve lossless performance (Guo et al., 2024; Li et al., 2024).

However, there is still a long way to go before DD can be practically used in real-world applications, *i.e.*, images in complex scenarios are characterized by significant variations in object sizes and the presence of a large amount of class-irrelevant information. To show that current DD methods fail to achieve satisfying performance in complex scenarios, we apply trajectory matching (Guo et al., 2024) on a 10-class subset from ImageNet-1K extracted by selecting similar classes of CIFAR-10, called IN1K-CIFAR-10. As depicted in Figure 1a, under three compressing ratios, DD's performances[1] on IN1K-CIFAR-10 are consistently worse than those on CIFAR-10.

To figure out the reason behind the above results, we take a closer look at the ImageNet-1K and CIFAR-10 from the data perspective. One key observation is that the percentage of discriminative features in the complex scenario, which can be visualized by Grad-CAM (Selvaraju et al., 2016), is much lower. From Figure 1b, CIFAR-10 images are mostly sticker-like, and activation maps have higher means and larger highly activated areas. By contrast, activation maps of the IN1K-CIFAR-10 subset exhibit much lower activation means and smaller highly activated areas. Previous methods (Du et al., 2022; Khaki et al., 2024) treat all pixels of synthetic images **equally**. Therefore, when applying these methods to more complex scenarios, the large ratio of low-activation areas leads to non-discriminative features dominating the learning process, resulting in a drop in performance.

From the supervision perspective, taking trajectory matching as an example, we investigate the impact of trajectories with different loss values on synthetic images. Specifically, we compare the trajectory-matching performance between i) using trajectory parameters with low losses only and ii) using trajectory parameters with high losses only. The effect on a single image is shown in Figure 2a. Low-loss supervision reduces the mean of Grad-CAM activation maps and shrinks the high-activation area (also shifted). By contrast, high-loss supervision increases the activation mean and expands the high-activation region.

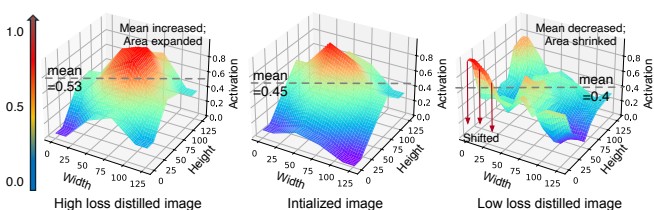

(a) High-loss supervision increases activation means and expands the high-activation area, while low-loss supervision reduces the activation mean and shifts to the wrong discriminative area.

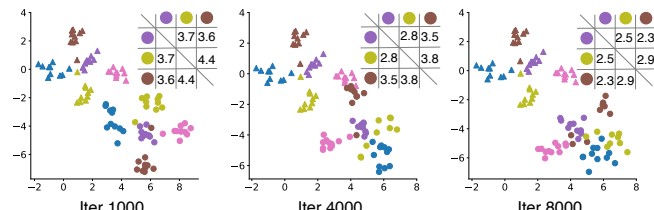

(b) Triangles and circles represent real and synthetic image features, respectively. As the distillation with low-loss only proceeds, more and more common patterns are introduced to synthetic images.

Additionally, we visualize the inter-class feature distribution for a broader view. In Figure 2b, we show the t-SNE of features of synthetic images distilled by only low-loss trajectories. As the distillation proceeds, synthetic image features of different classes continuously come closer, and the confusion among classes becomes more severe, which is likely caused by common patterns. The above two phenomenons confirm that low-loss supervision primarily reduces the representation of discriminative features and embeds more common patterns into synthetic images, harming DD's performance.

Figure 2: **(a)** Grad-CAM activation maps of the image with initialization, high-loss supervision distillation, and low-loss supervision distillation. **(b)** t-SNE visualization of image features with **only** low-loss supervision. Different colors represent different classes. The top right is inter-class distance computed by the average of point-wise distances.

Based on the above observations, we propose to **Emphasize Discriminative Features** (EDF), built on trajectory matching. To synthesize more discriminative features in the distilled data, we enable discriminative areas to receive more updates compared with non-discriminative ones. This is achieved by guiding the optimization of synthetic images with gradient weights computed from Grad-CAM activation maps. Highly activated pixels are assigned higher gradients for enhancement. To mitigate

---

[1]We compare recovery ratios because datasets are different.

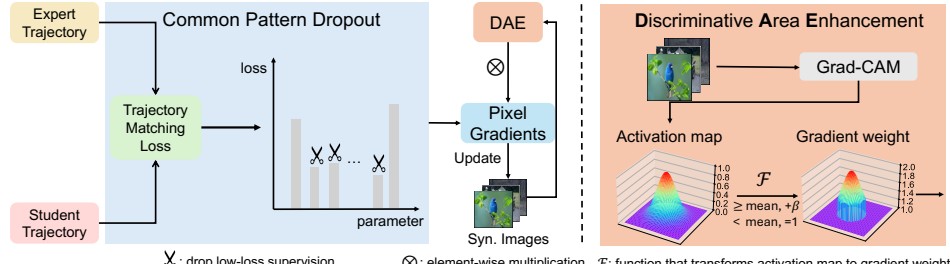

Figure 3: Workflow of **Emphasize Discriminative Features** (EDF). EDF comprises two modules: (1) *Common Pattern Dropout*, which filters out low-loss signals, and the (2) *Discriminative Area Enhancement*, which amplifies gradients in critical regions.

the negative impact of common patterns, the EDF distinguishes between different supervision signals by dropping those with a low trajectory matching loss according to a drop ratio.

To help the community explore DD in complex scenarios, we extract new subsets from ImageNet-1K with various levels of complexity and build the Complex DD benchmark (Comp-DD). The complexity levels of these new subsets are determined by the average ratios of high-activation areas (Grad-CAM activation value > predefined threshold). We run EDF and several typical DD methods on partial Comp-DD and will release the full benchmark for future studies to further improve performance.

In summary, EDF consistently achieves state-of-the-art (SOTA) performance across various datasets, underscoring the effectiveness of emphasizing discriminative features. On several ImageNet-1K subsets, EDF achieves lossless performance. To the best of our knowledge, we are the first to achieve lossless performance on ImageNet-1K subsets. We build the Complex Dataset Distillation benchmark based on complexity, providing convenience for future research to continue improving DD's performance in complex scenarios.

## 2 METHOD

Our approach, **Emphasize Discriminative Features** (EDF), enhances discriminative features in synthetic images during distillation. As shown in Figure 3, EDF first trains trajectories on real images $T$ and synthetic images $S$ and computes the trajectory matching loss. Then, *Common Pattern Dropout* filters out low-loss supervision signals, retaining high-loss ones for backpropagation. After obtaining gradients for the synthetic images, *Discriminative Area Enhancement* uses dynamically extracted Grad-CAM activation maps to rescale pixel gradients, focusing updates on discriminative regions.

### 2.1 COMMON PATTERN DROPOUT

This module reduces common patterns in supervision by matching expert and student trajectories on real and synthetic data, then removing low-loss elements. This ensures only meaningful supervision enhances the model's ability to capture discriminative features.

**Trajectory Generation and Loss Computation.** To generate expert and student trajectories, we first train agent models on real data for $E$ epochs, saving the resulting parameters as expert trajectories, denoted by $\{\theta_t\}_0^E$. At each distillation iteration, we randomly select an initial point $\theta_t$ and a target point $\theta_{t+M}$ from these expert trajectories. Similarly, student trajectories are produced by initializing an agent model at $\theta_t$ and training it on the synthetic dataset, yielding the parameters $\{\hat{\theta}_t\}_0^N$. The trajectory matching loss is then computed by comparing the final student parameters $\hat{\theta}_{t+N}$ with the expert's target parameters $\theta_{t+M}$, normalized by the initial difference:

$$L = \frac{||\hat{\theta}_{t+N} - \theta_{t+M}||^2}{||\theta_{t+M} - \theta_t||^2}. \tag{1}$$

Instead of directly summing this loss, we decompose it into an array of individual losses between corresponding parameters in the expert and student trajectories, represented as $L = \{l_1, l_2, \ldots, l_P\}$, where $P$ is the number of parameters, and $l_i$ is the loss associated with the $i$-th parameter.

**Low-loss Element Dropping.** Our analyses of Figure 2a and 2b show that low-loss signals typically correspond to common patterns, which hinder the learning of key discriminative features, particularly in complex scenarios. To address this, we sort the array of losses computed from the previous step in ascending order. Using a predefined dropout ratio $\alpha$, we discard the smallest $\lfloor \alpha \cdot P \rfloor$ losses ($\lfloor \rfloor$ denotes the floor function), which are assumed to capture common, non-discriminative features. The remaining losses are summed and normalized to form the final supervision:

$$L \xrightarrow{\text{sort}} L' = \{\underbrace{l_1, l_2, \cdots, l_{\lfloor \alpha \cdot P \rfloor}}_{\text{dropout}}, \underbrace{l_{\lfloor \alpha \cdot P \rfloor + 1}, \cdots, l_P}_{\text{sum\&normalize}}\}, \tag{2}$$

where $L'$ represents the updated loss array after dropping the lowest $\lfloor \alpha \cdot P \rfloor$ elements. The remaining losses, $l_{\lfloor \alpha \cdot P \rfloor + 1}, \ldots, l_P$, are then summed and normalized to form the final supervision signal.

## 2.2 DISCRIMINATIVE AREA ENHANCEMENT

After the pruned loss from *Common Pattern Dropout* is backpropagated, this module amplifies the importance of discriminative regions in synthetic images. Grad-CAM activation maps are dynamically extracted from the synthetic data to highlight areas most relevant for classification. These activation maps are then used to rescale the pixel gradients, applying a weighted update that prioritizes highly activated regions, thereby focusing the learning process on key discriminative features.

**Activation Map Extraction.** Grad-CAM generates class-specific activation maps by leveraging the gradients that flow into the final convolutional layer, highlighting key areas relevant for predicting a target class. To compute these maps, we first train a convolutional model $\mathcal{G}$ on the real dataset. Following the Grad-CAM formulation (Equation 3), we calculate the activation map for each class $c$: $M^c \in \mathbb{R}^{IPC \times H \times W}$ on the synthetic images (IPC is the number of images per class). The activation map $M^c$ is a gradient-weighted sum of feature maps across all convolutional layers:

$$\alpha^c = \frac{1}{Z} \sum_h \sum_w \frac{\partial y^c}{\partial A_{h,w}^l} \qquad M^c = ReLU(\sum_l \alpha_l^c A^l), \tag{3}$$

$\alpha^c$ represents the weight of the activation of the $l$-th convolutional layer, $A^l$, computed by gradients. Finally, we concatenate $M^c$ of all images in class $c$ and obtain $M \in \mathbb{R}^{|S| \times H \times W}$.

**Discriminative Area Biased Update.** A major limitation of previous DD algorithms (Cazenavette et al., 2022; Du et al., 2022; Guo et al., 2024) on the complex scenario is that they treat each pixel **equally** and provide no guidance for the distillation process on which area of synthetic images should be emphasized. Therefore, we propose to update synthetic images in a biased manner. Instead of treating each pixel equally, we enhance the significance of discriminative areas by guiding the optimization with activation maps extracted in the previous step. We define the discriminative area of a synthetic image as the percentage of pixels with activation values above the mean since synthetic images are dynamically changing (see Section 4.3 for discussion). Specifically, we process activation maps from the previous step with a function $\mathcal{F}(M, \beta)$ to create weights for pixel gradients as follows:

$$\mathcal{F}(M_{h,w}^i, \beta) = \begin{cases} 1 & \text{if } M_{h,w}^i < \bar{M}^i, \\ \beta + M_{h,w}^i & \text{if } M_{h,w}^i \geq \bar{M}^i. \end{cases} \tag{4}$$

$M_{h,w}^i$ denotes the activation value of the $i$ the image at coordinate $(h, w)$, and $\bar{M}^i$ denotes the mean activation of $M^i$. $\beta \geq 1$ is called the *enhancement factor*. Then, we rescale the gradient matrix of synthetic images by multiplying it with the weight matrix element-wise:

$$(\nabla D_{syn})_{edf} = \nabla D_{syn} \circ \mathcal{F}(M, \beta). \tag{5}$$

We drag gradients of discriminative areas to a higher range so that they receive more updates.

**Dynamic Update of Activation Maps.** As synthetic images are optimized, high-activation regions shift over time. To capture these changes, we recompute the activation maps every $K$ iterations, focusing updates on the most relevant areas. The frequency $K$ is a tunable hyperparameter, adjusted based on the learning rate of the synthetic images (see Section 4.3 for details). This ensures evolving discriminative areas are accurately captured. The complete algorithm is provided in the appendix A.

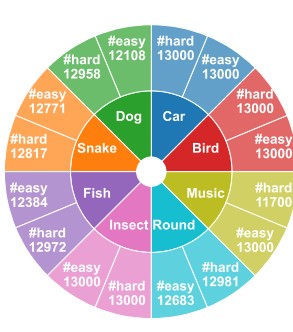 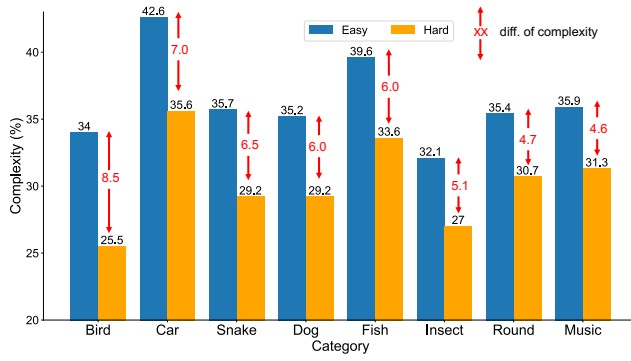

(a) Number of images in each subset  (b) Complexity of Easy vs. Hard subsets in each category

Figure 4: **(a)** Statistics of the training set in the Comp-DD benchmark. Each subset contains 500 images in the validation set. **(b)** Comparison of subset-level complexity between easy and hard subsets across all categories. The complexity of hard subsets is higher than that of easy subsets.

## 3 COMPLEX DD BENCHMARK

We introduce the **Comp**lex **D**ataset **D**istillation (Comp-DD) benchmark, which is constructed by selecting subsets from ImageNet-1K based on their complexity. This benchmark represents an early and pioneering effort to address dataset distillation in complex scenarios. Although there are numerous benchmarks (Krizhevsky, 2009; Le & Yang, 2015; Cui et al., 2022b) for simpler tasks, there is a notable absence of benchmarks designed specifically for complex scenarios. This gap presents a significant challenge to advancing research in this area and limits the practical application of dataset distillation. To bridge this gap, we propose the first dataset distillation benchmark explicitly built around scenario complexity, aiming to promote further exploration within the DD community.

**Complexity Metrics.** We evaluate the complexity of an image by measuring the average size of high-activation regions of the Grad-CAM activation map. Using a pre-trained ResNet model, we first generate Grad-CAM activation maps for all images, class by class. For each image, we calculate the percentage of pixels with activation values above a predefined threshold (set to 0.5 in our case), with higher percentages indicating lower complexity (more clarifications can be found in Appendix D.2). Formally, the complexity of the $i$-th image is computed as $\sum_h \sum_w \mathbb{1}[M_{h,w}^i \geq 0.5]$ where $\mathbb{1}$ is the indicator function. The complexity of each class is then determined by averaging the complexity scores across all images within that class.

**Subset Selection.** To reduce the influence of class differences, we select subsets from each *category*, where a category consists of classes representing visually similar objects or animals of the same species. This approach allows us to focus on complexity while controlling for inter-class variability.

Specifically, we first manually identify representative categories in ImageNet-1K with sufficient numbers of classes ($\geq 20$). For each category, we rank the classes by complexity in descending order. Following established practice, we construct two ten-class subsets for each category: the *easy* subset, comprising the ten least complex classes, and the *hard* subset, comprising the ten most complex classes. The subset-level complexity is determined by averaging the complexity scores across all classes in each subset.

**Statistics.** We carefully selected eight categories from ImageNet-1K: Bird, Car, Dog, Fish, Snake, Insect, Round, and Music. Each category contains two ten-class subsets: one *easy* and one *hard*, with difficulty determined by the complexity metrics outlined above. Figure 4a summarizes the number of training images in each subset, while all subsets contain 500 images in the validation set. To illustrate the difference between easy and hard subsets, Figure 4b compares the subset-level complexity for each category. As expected, the hard subsets display significantly higher complexity than the easy subsets. For a detailed breakdown of the classes in each subset, please refer to Appendix D.1.

| Dataset | IPC | DD | | | | | Eval. w/ Knowledge Distillation | | | Full |
|---|---|---|---|---|---|---|---|---|---|---|
| | | Random | MTT | FTD | DATM | EDF | SRe2L | RDED | EDF | |
| ImageNette | 1 | 12.6±1.5 | 47.7±0.9 | 52.2±1.0 | 52.5±1.0 | **52.6±0.5** | 20.8±0.2 | **28.9±0.1** | 25.7±0.4 | |
| | 10 | 44.8±1.3 | 63.0±1.3 | 67.7±0.7 | 68.9±0.8 | **71.0±0.8** | 50.6±0.8 | 59.0±1.0 | **64.5±0.6** | 87.8±1.0 |
| | 50 | 60.4±1.4 | 🖼 | 🖼 | 75.4±0.9 | **77.8±0.5** | 73.8±0.6 | 83.1±0.6 | **84.8±0.5** | |
| ImageWoof | 1 | 11.4±1.3 | 28.6±0.8 | 30.1±1.0 | 30.4±0.7 | **30.8±1.0** | 15.8±0.8 | 18.0±0.3 | **19.2±0.2** | |
| | 10 | 20.2±1.2 | 35.8±1.8 | 38.8±1.4 | 40.5±0.6 | **41.8±0.2** | 38.4±0.4 | 40.1±0.2 | **42.3±0.3** | 66.5±1.3 |
| | 50 | 28.2±0.9 | 🖼 | 🖼 | 47.1±1.1 | **48.4±0.5** | 49.2±0.4 | 60.8±0.5 | **61.6±0.8** | |
| ImageMeow | 1 | 11.2±1.2 | 30.7±1.6 | 33.8±1.5 | 34.0±0.5 | **34.5±0.2** | **22.2±0.6** | 19.2±0.8 | 20.8±0.5 | |
| | 10 | 22.4±0.8 | 40.4±2.2 | 43.3±0.6 | 48.9±1.1 | **52.6±0.4** | 27.4±0.5 | 44.2±0.6 | **48.4±0.7** | 65.2±0.8 |
| | 50 | 38.0±0.5 | 🖼 | 🖼 | 56.4±0.9 | **59.5±0.6** | 35.8±0.7 | 55.0±0.6 | **58.2±0.9** | |
| ImageYellow | 1 | 14.8±1.0 | 45.2±0.8 | 47.7±1.1 | 48.5±0.4 | **49.4±0.5** | 31.8±0.7 | 30.6±0.2 | **33.5±0.6** | |
| | 10 | 41.8±1.1 | 60.0±1.5 | 62.8±1.4 | 65.1±0.7 | **68.2±0.4** | 48.2±0.5 | 59.2±0.5 | **60.8±0.5** | 83.2±0.9 |
| | 50 | 54.6±0.5 | 🖼 | 🖼 | 70.5±0.8 | **73.6±0.8** | 57.6±0.9 | 75.8±0.7 | **76.2±0.3** | |
| ImageFruit | 1 | 12.4±0.9 | 26.6±0.8 | 29.1±0.9 | 30.9±1.0 | **32.8±0.6** | 23.4±0.5 | **33.8±0.4** | 29.6±0.4 | |
| | 10 | 20.0±0.6 | 40.3±0.5 | 44.9±1.5 | 45.5±0.9 | **46.2±0.6** | 39.2±0.7 | 45.4±0.6 | **48.4±0.8** | 64.4±0.8 |
| | 50 | 33.6±0.9 | 🖼 | 🖼 | 48.2±0.5 | **50.5±0.5** | 44.2±0.8 | 54.8±0.9 | **56.4±0.6** | |
| ImageSquawk | 1 | 13.2±1.1 | 39.4±1.5 | 40.5±0.9 | 41.1±0.6 | **41.8±0.5** | 21.2±1.0 | **33.8±0.6** | 30.5±0.5 | |
| | 10 | 29.6±1.5 | 52.3±1.0 | 58.4±1.5 | 61.8±1.3 | **65.4±0.8** | 39.2±0.3 | 59.0±0.5 | **59.4±0.6** | 86.4±0.8 |
| | 50 | 52.8±0.4 | 🖼 | 🖼 | 71.0±1.2 | **74.8±1.2** | 56.8±0.4 | 77.2±1.2 | **77.8±0.5** | |

Table 1: Results of depth-5 ConvNet on ImageNet-1K subsets. 🖼 indicates worse performance than DATM. EDF achieves SOTAs on all settings compared with DD methods. Compared with SRe2L and RDED, we achieve SOTAs on 14 out of 18 settings.

# 4 EXPERIMENT

## 4.1 EXPERIMENTAL SETUP

**Datasets and Architecture.** We conduct a comprehensive evaluation of EDF on six well-known subsets (Howard, 2019) of ImageNet-1K: ImageNette, ImageWoof, ImageMeow, ImageYellow, ImageFruit, and ImageSquawk. Each subset contains ten classes, with approximately 13,000 images in the training set and 500 images in the validation set. On the Comp-DD benchmark, we report the results of the Bird, Car, and Dog categories. For all experiments, we use a 5-layer ConvNet (ConvNetD5) as both the distillation and the evaluation architecture. For cross-architecture evaluation (see results in Appendix C.1), we validate synthetic data accuracy on ResNet-18 (He et al., 2015), VGG11 (Simonyan & Zisserman, 2014), and Alexnet (Krizhevsky et al., 2012).

**Baselines.** We compare two baselines: dataset distillation (DD) methods and methods utilizing knowledge distillation (Eval. w/ Knowledge Distillation) (Hinton et al., 2015). For DD methods, we include trajectory-matching-based methods such as MTT (Cazenavette et al., 2022), FTD (Du et al., 2022), and DATM (Guo et al., 2024). In the knowledge distillation group, we compare against SRe2L (Yin et al., 2023) and RDED (Sun et al., 2023). The results for subsets not covered in these papers are obtained through replication using the official open-source codebases and hyperparameters.

## 4.2 MAIN RESULTS

**ImageNet-1K Subsets.** We mainly conduct extensive experiments on the ImageNet-1K subsets to compare the performance of EDF with other approaches. The detailed results are shown in Table 1. EDF consistently achieves state-of-the-art (SOTA) results across all settings when compared to other dataset distillation methods. On larger IPCs, *i.e.*, 200 or 300, the performance of EDF significantly outperforms that observed with smaller IPCs. We achieve lossless performances on ImageMeow and ImageYellow under IPC300, 23% of real data, as shown in Table 2.

Table 2: Lossless performance under IPC300.

| Subset | ImageMeow | | ImageYellow | |
|---|---|---|---|---|
| IPC | 200 | 300 | 200 | 300 |
| Random | 52.8±0.4 | 55.3±0.3 | 70.5±0.5 | 72.8±0.8 |
| EDF | 62.5±0.7 | **65.9±0.6** | 81.0±0.9 | **84.2±0.7** |
| Full | 65.2±1.3 | | 83.2±0.9 | |

When evaluated against Eval. w/ Knowledge Distillation methods, our distilled datasets outperform SRe2L and RDED in 14 out of 18 settings. It is important to note that applying knowledge distillation (KD) for evaluation tends to reduce EDF's pure dataset distillation performance, particularly in low IPC (images per class) settings such as IPC1 and IPC10. This occurs because smaller IPCs lack the capacity to effectively incorporate the knowledge from a well-trained teacher model. We also provide results without knowledge distillation in Appendix C.2.

| Method | Bird-Easy | | Bird-Hard | | Dog-Easy | | Dog-Hard | | Car-Easy | | Car-Hard | |
|---|---|---|---|---|---|---|---|---|---|---|---|---|
| IPC | 10 | 50 | 10 | 50 | 10 | 50 | 10 | 50 | 10 | 50 | 10 | 50 |
| Random | 32.4±0.5 | 53.8±0.6 | 22.6±0.7 | 41.8±0.5 | 26.0±0.4 | 30.8±0.8 | 14.5±0.2 | 27.6±0.7 | 18.2±0.4 | 34.4±0.3 | 25.6±0.5 | 40.4±0.5 |
| FTD | 60.0±1.1 | 63.4±0.6 | 54.4±0.8 | 59.6±1.2 | 41.1±1.3 | 45.9±0.9 | 36.5±1.1 | 43.5±0.9 | 44.4±1.1 | 49.6±0.5 | 52.1±0.5 | 55.6±0.9 |
| DATM | 62.2±0.4 | 67.1±0.3 | 56.0±0.5 | 62.9±0.8 | 42.8±0.7 | 48.2±0.5 | 38.6±0.7 | 47.4±0.5 | 46.4±0.5 | 53.8±0.6 | 53.2±0.6 | 58.7±0.8 |
| EDF | **63.4±0.5** | **69.0±0.8** | **57.1±0.4** | **64.8±0.6** | **43.2±0.5** | **49.4±0.8** | **39.6±0.9** | **49.2±0.3** | **47.6±0.7** | **54.6±0.2** | **55.4±0.8** | **61.0±0.5** |
| Full | 81.6±1.0 | | 82.4±0.8 | | 57.3±0.3 | | 58.4±0.5 | | 63.5±0.2 | | 72.8±1.1 | |

(a) EDF achieves SOTAs on Bird, Dog, and Car categories of the Comp-DD benchmark

| Category | Bird | | | | Dog | | | | Car | | | |
|---|---|---|---|---|---|---|---|---|---|---|---|---|
| IPC | 10 | | 50 | | 10 | | 50 | | 10 | | 50 | |
| Complexity | Easy | Hard | Easy | Hard | Easy | Hard | Easy | Hard | Easy | Hard | Easy | Hard |
| Recovery ratio (%) | **77.7** | 69.3 | **84.6** | 76.9 | **75.4** | 67.8 | **86.2** | 84.2 | **76.2** | 75.6 | **87.4** | 83.7 |

(b) Recovery ratios of easy subsets are **higher** than that of hard subsets, aligning with the complexity metrics.

Table 3: **(a)** Partial results on Bird, Dog, and Car categories of the Complex DD Benchmark under IPC 10 and 50. **(b)** Recovery ratios (RR) of EDF on the partial Complex DD Benchmark.

**Comp-DD Benchmark.** The results for EDF on the Bird, Car, and Dog categories from the Comp-DD Benchmark are shown in Table 3. EDF demonstrates superior test accuracies and recovery ratios across both easy and hard subsets. As expected, the recovery ratios for easy subsets are consistently higher than those for hard subsets, confirming that the hard subsets present a greater challenge for dataset distillation methods. These results validate our complexity metrics, which effectively distinguish the varying levels of difficulty between easy and hard subsets.

## 4.3 ABLATION STUDY

We conduct an ablation study to evaluate the impact of EDF's key components, including the supervision dropout ratio, strategies for discriminative area enhancement, and the frequency of activation map updates. Unless otherwise specified, the following results are all based on ConvNetD5.

**Effect of Modules.** EDF introduces two key modules: **Discriminative Area Enhancement** (DAE) and **Common Pattern Dropout** (CPD). We conduct an ablation study to assess the contribution of each module independently. The results, presented in Table 4, demonstrate that both DAE and CPD significantly improve the baseline performance. DAE's biased updates toward high-activation areas using activation-based gradient weights effectively enhance the discriminative features in synthetic images. CPD, on the other hand, mitigates the negative influence of common patterns by filtering out low-loss supervision, ensuring that the synthetic images retain their discriminative properties.

| DAE | CPD | Accuracy(%) |
|---|---|---|
| | | 39.2 |
| | ✓ | 40.3 |
| ✓ | | 41.1 |
| ✓ | ✓ | 41.8 |

(a) ImageWoof, IPC10

| DAE | CPD | Accuracy(%) |
|---|---|---|
| | | 48.9 |
| | ✓ | 49.5 |
| ✓ | | 51.2 |
| ✓ | ✓ | 52.6 |

(b) ImageMeow, IPC10

| DAE | CPD | Accuracy(%) |
|---|---|---|
| | | 65.1 |
| | ✓ | 66.2 |
| ✓ | | 67.5 |
| ✓ | ✓ | 68.2 |

(c) ImageYellow, IPC10

Table 4: Ablation results of two modules, DAE and CPD, on three ImageNet-1K subsets. Both modules bring improvements to the performance, underscoring individual efficacy.

**Supervision Dropout Ratio.** The dropout ratio in CPD is critical for balancing the removal of common patterns and dataset capacity (IPC). As shown in Table 5a, smaller IPCs benefit most from moderate dropout ratios (12.5-25%), which filter low-loss signals while preserving important

| Ratio (%) | | 0 | 12.5 | 25 | 37.5 | 50 | 75 |
|---|---|---|---|---|---|---|---|
| | 1 | **32.8** | 32.4 | 32.3 | 31.8 | 30.6 | 29.1 |
| ImageFruit | 10 | 45.4 | 45.9 | **46.5** | 46.2 | 45.8 | 44.3 |
| | 50 | 49.5 | 50.1 | 50.7 | **50.9** | 50.6 | 49.2 |
| | 1 | **41.8** | 41.3 | 41.2 | 41.0 | 39.6 | 38.1 |
| ImageSquawk | 10 | 64.8 | 65.0 | **65.4** | 65.2 | 64.9 | 63.2 |
| | 50 | 73.9 | 74.2 | 74.6 | **74.8** | 74.5 | 72.8 |

| Frequency (iter.) | | 1 | 50 | 100 | 200 |
|---|---|---|---|---|---|
| | 1 | 49.4 | **51.2** | 50.5 | 49.5 |
| ImageNette | 10 | 68.4 | 69.8 | **71.0** | 70.6 |
| | 50 | 72.5 | 75.6 | 76.5 | **77.8** |
| | 1 | 47.8 | **49.4** | 49.2 | 48.2 |
| ImageYellow | 10 | 66.4 | 67.8 | **68.2** | 67.2 |
| | 50 | 70.4 | 72.2 | 73.1 | **73.6** |

(a) Within a reasonable range, the target supervision dropout ratio increases as the IPC becomes larger. Dropping too much supervision could result in losing too much information.

(b) Within a reasonable range, a higher frequency performs better on small IPCs, while larger IPCs prefer a lower frequency.

Table 5: **(a)** Results of different supervision dropout ratios across various IPCs. **(b)** Results of different activation map update frequencies across various IPCs.

| IPC | Enhancement Factor ($\beta$) | | | | |
|---|---|---|---|---|---|
| | 0.5 | 1 | 2 | 5 | 10 |
| 1 | 33.4 | **34.5** | 34.3 | 33.2 | 31.8 |
| 10 | 50.1 | **52.6** | 52.1 | 49.4 | 49.0 |
| 50 | 57.8 | **59.5** | 59.2 | 58.1 | 57.6 |

| IPC | Enhancement Factor ($\beta$) | | | | |
|---|---|---|---|---|---|
| | 0.5 | 1 | 2 | 5 | 10 |
| 1 | 29.1 | **30.8** | 30.5 | 30.2 | 28.8 |
| 10 | 40.9 | 41.2 | **41.8** | 41.0 | 40.4 |
| 50 | 47.5 | 48.2 | **48.4** | 48.1 | 47.2 |

| IPC | Activation Threshold | | | |
|---|---|---|---|---|
| | 0.2 | 0.5 | 0.8 | mean |
| 1 | 34.2 | 34.0 | 33.8 | **34.5** |
| 10 | 51.2 | 52.3 | 51.5 | **52.6** |
| 50 | 58.0 | 59.0 | 58.4 | **59.5** |

(a) Results on ImageMeow (left) and ImageWoof (right). ImageWoof has a higher complexity. Enhancement factor should be set within a reasonable range ($\geq 1$ and $\leq 5$ in general).

(b) Using "mean" as a dynamic threshold gives the best performance on three IPCs.

Table 6: **(a)** Ablation of the enhancement factor on ImageMeow and ImageWoof, both IPC10. **(b)** Ablation of the activation threshold on ImageMeow IPC10.

information. For larger IPCs, higher dropout ratios (37.5-50%) improve performance, as these datasets can tolerate more aggressive filtering. However, an excessively high ratio (e.g., 75%) reduces performance across all IPCs by discarding too much information, weakening the ability to learn.

**Frequency of Activation Map Update.** To accurately capture the evolving discriminative features in synthetic images, EDF dynamically updates the Grad-CAM activation maps at a predefined frequency. The choice of update frequency should be adjusted based on the IPC to achieve optimal performance. As shown in Table 5b, larger IPCs benefit from a lower update frequency, as the pixel learning rate is set lower for more stable distillation. In contrast, smaller IPCs require a higher update frequency to effectively adapt to the faster changes in the synthetic images during training.

This trend is influenced by the pixel learning rate: larger IPCs can use lower rates to ensure smooth convergence, making frequent updates unnecessary. Smaller IPCs, with limited data capacity, require higher learning rates and more frequent updates to quickly adapt to changes in discriminative areas. Thus, selecting the appropriate update frequency is essential for balancing stability and adaptability in the distillation process, depending on dataset size and complexity.

**Strategies for Discriminative Area Enhancement.** The Discriminative Area Enhancement (DAE) component involves two key factors: the enhancement factor $\beta$ and the threshold for activation maps. Ablation studies (Table 6a) show that the best performance is achieved when $\beta$ is between 1 and 2. When $\beta < 1$, some discriminative areas are diminished rather than enhanced, as their gradient weights become $< 1$. Conversely, excessively large $\beta$ values ($\geq 10$) lead to overemphasis on certain areas, distorting the overall learning process (see Appendix C.3 for examples of this distortion). Therefore, $\beta$ should be reasonably controlled to balance the emphasis on discriminative regions.

Regarding the threshold for activation maps, using the mean activation value as a dynamic threshold results in better performance compared to using a fixed threshold. This is because the mean adapts to the evolving activation maps during training, whereas a fixed threshold risks either emphasizing low-activation areas if set too low or omitting key discriminative features if set too high.

## 5 ANALYSIS AND DISCUSSION

**Disitlled Images of Different Supervision.** As pointed out earlier, low-loss supervision tends to introduce common patterns, such as backgrounds and general colors, while high-loss supervision contains discriminative, class-specific features. To visualize this effect, we select two images with

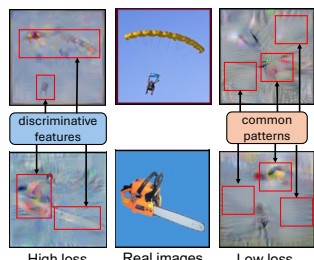 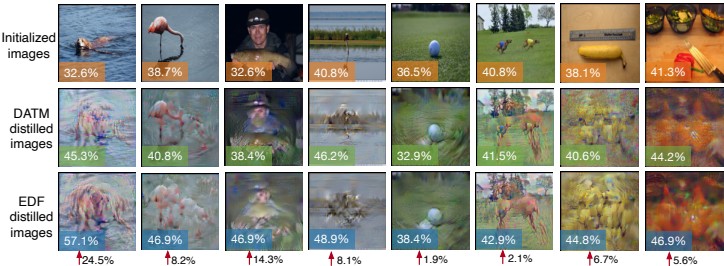

(a) Low-loss supervision mainly embeds common patterns (background, colors).

(b) EDF largely increases the percentage of discriminative areas (bottom left figure of each image) with an average of 9%, achieving the highest. Our distilled images contain more discriminative features.

Figure 5: **(a)** Comparison of **discriminative areas** in images produced by initialization, DATM, and EDF. Figures at the bottom are increments made by EDF over the initial image. **(b)** Comparison between high-loss and low-loss supervision distilled images.

similar backgrounds and colors, but distinct objects. Two images are then distilled by high-loss and low-loss supervision, respectively. As shown in Figure 5a, common patterns are indeed widely present in low-loss supervision distilled images, making two images hard to distinguish. In contrast, high-loss supervision preserves more discriminative details, enabling the model to distinguish between two classes. This further confirms the validity of dropping low-loss supervision and underscores the effectiveness of the *Common Pattern Dropout* (CPD) module in mitigating the negative impact of common features.

**Enhancement of Discriminative Areas.** Our Discriminative Area Enhancement (DAE) module aims to amplify updates in high-activation areas of synthetic images, as identified by Grad-CAM. To show how DAE enhances discriminative areas, we visualize the same group of images under initialization, DATM distillation, and EDF distillation in Figure 5b We also report discriminative area statistics, computed by the percentage of pixels whose activation values are higher than the mean, on each image at the bottom left. As can be discovered, DATM is capable of increasing discriminative regions, while EDF can achieve a more significant enhancement. Visually, the enhancement manifests through an increased number of core objects and enlarged areas of class-specific features. Moreover, EDF's enhancement is more pronounced especially when the image has smaller discriminative areas initially, e.g. discriminative features of the first column image increase by 24.5%. These phenomena demonstrate the effectiveness of EDF in capturing and emphasizing discriminative features.

**Supervision Dropout Criteria.** To assess the effectiveness of supervision dropout strategies, we compare several dropout approaches. These strategies are classified into two categories: (i) dynamic dropout, which includes random selection from all layers, and (ii) static dropout, which includes uniform selection across layers and fixed selection from the first, middle, or last layers. As shown in Table 7, all strategies except EDF's loss-based dropout lead to performance degradation, with uniform selection and last-layer dropout causing the most significant performance loss.

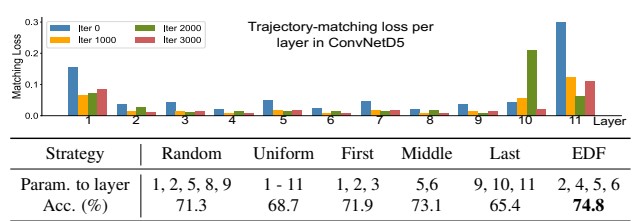

Table 7: EDF's loss-wise dropout performs the best. The dropping ratio of all criteria is fixed at 25%. "Param. to layer" refers to layers that contain dropped trajectory parameters.

| Strategy | Random | Uniform | First | Middle | Last | EDF |
|---|---|---|---|---|---|---|
| Param. to layer | 1, 2, 5, 8, 9 | 1 - 11 | 1, 2, 3 | 5,6 | 9, 10, 11 | 2, 4, 5, 6 |
| Acc. (%) | 71.3 | 68.7 | 71.9 | 73.1 | 65.4 | **74.8** |

The reasons for this are twofold. First, low-loss trajectory parameters—primarily located in the shallow layers of the model—are the main source of common patterns. Discarding supervision from deep layers, where loss values are higher (random selection, uniform selection, or last-layer dropout), reduces the presence of discriminative features. Second, static dropout fails to account for the dynamic nature of low-loss supervision, as trajectory-matching losses vary across layers as the

distillation process evolves. By addressing these issues, EDF's loss-based dropout in CPD mitigates the effects of common patterns and yields superior performance.

## 6    RELATED WORK

**Approaches.** Dataset Distillation (DD) aims to create compact datasets that maintain performance levels comparable to full-scale datasets. It can be applied in practical fields such as continual learning (Masarczyk & Tautkute, 2020; Rosasco et al., 2021), privacy preservation (Dong et al., 2022; Yu et al., 2023), and neural architecture search (Jin et al., 2018; Pasunuru & Bansal, 2019). Approaches in DD can be categorized into two primary approaches: matching-based and knowledge-distillation-based.

1) *Matching-based approaches* are foundational in DD research, focusing on aligning synthetic data with real datasets by capturing essential patterns. Landmark works like gradient matching (DC) (Zhao et al., 2021), distribution matching (DM) (Zhao & Bilen, 2021a), and trajectory matching (MTT) (Cazenavette et al., 2022) extract critical metrics from real datasets, then replicate these metrics in synthetic data. Subsequent research has refined these methods, improving the fidelity of distilled datasets (Zhao & Bilen, 2021b; Wang et al., 2022; Zhao et al., 2023; Lee et al., 2022b; Liu et al., 2023a;b; Cazenavette et al., 2023; Sajedi et al., 2023; Khaki et al., 2024). Data selection techniques have been integrated to synthesize more representative samples (Xu et al., 2023; Sundar et al., 2023; Lee & Chung, 2024). Recent advancements optimize distillation for different image-per-class (IPC) settings, balancing dataset size and information retention (Du et al., 2023; Chen et al., 2023; Guo et al., 2024; Li et al., 2024; Lee & Chung, 2024). Moreover, soft labels have been widely applied to improve the performance (Sucholutsky & Schonlau, 2021; Cui et al., 2022a; Qin et al., 2024; Yu et al., 2024). Despite these improvements, most matching-based approaches treat all pixels uniformly, failing to emphasize discriminative regions and often overlooking distinctions between supervision signals, limiting their effectiveness on complex datasets like ImageNet-1K.

2) *Knowledge-distillation-based approaches* take an alternative route by aligning teacher-student model outputs when evaluating distilled datasets. Notable examples include SRe2L (Yin et al., 2023) and RDED (Sun et al., 2023), where the student model is trained by aligning outputs with outputs of a teacher model on the same batch of synthetic data, specifically by minimizing the Kullback-Leibler (KL) divergence between the student's predictions and the teacher's output. In our work, we adopt knowledge distillation as a validation strategy for fair comparisons.

**Benchmarks.** DD research has mainly focused on simpler datasets such as CIFAR (Krizhevsky, 2009), TinyImageNet (Le & Yang, 2015), and DC-BENCH (Cui et al., 2022b). These datasets contain a high proportion of class-specific information, enabling DD methods to extract and synthesize discriminative features more easily. However, research in more complex scenarios has been limited. To address this, we propose the Comp-DD benchmark, which systematically explores dataset distillation complexity by curating subsets from ImageNet-1K with varying degrees of difficulty. This benchmark provides a more rigorous evaluation framework, facilitating deeper exploration of DD in complex, real-world settings and encouraging further advances in the field.

## 7    CONCLUSION

We introduced Emphasize Discriminative Features (EDF), a dataset distillation method that enhances class-specific regions in synthetic images. EDF addresses two key limitations of prior methods: i) enhancing discriminative regions in synthetic images using Grad-CAM activation maps, and ii) filtering out low-loss signals that embed common patterns through *Common Pattern Dropout (CPD)* and *Discriminative Area Enhancement (DAE)*. EDF achieves state-of-the-art results across ImageNet-1K subsets, including lossless performance on several of them. We also proposed the Comp-DD benchmark, designed to evaluate dataset distillation in both simple and complex settings.

**Limitations and Future Work.** EDF dynamically updates Grad-CAM activation maps of synthetic images according to an update frequency. This may introduce extra computation, especially when the IPC is large. Also, we only use Grad-CAM to evaluate discriminative areas of an image in this work. In the future, other indicators that can identify discriminative features of an image can be used jointly to include more perspectives.

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

APPENDIX

We organize our appendix as follows.

**Algorithm of EDF:**

**Experimental Settings:**

**Additional Experimental Results and Findings:**

**Comp-DD Benchmark**

**Visualization**

**Related Work**

## A    ALGORITHM OF EDF

Algorithm 1 provides a pseudo-code of EDF. Lines 1-7 specify inputs of the EDF, including a trajectory-matching algorithm $\mathcal{A}$, the model for Grad-CAM $\mathcal{G}$, the frequency of activation map update $K$, the supervision dropout ratio $\alpha$, the enhancement factor $\beta$, the activation map processing function $\mathcal{F}$, and the number of distillation iterations $T$.

Lines 12-14 describe the Common Pattern Dropout module. After we obtain the trajectory matching losses from $\mathcal{A}$, we sort them in ascending order to get ordered losses. Then, the smallest $\alpha|L|$ elements are dropped as they introduce non-discriminative common patterns.

Lines 15-19 describe the Discriminative Area Enhancement module. For every $K$ iterations, we update activation maps of synthetic images. The gradients of synthetic images are then processed by the function $\mathcal{F}$ (see Equation 4 for the computation). Finally, synthetic images are updated biasedly towards discriminative areas.

## B    EXPERIMENTAL SETTINGS

### B.1    TRAINING DETAILS

We follow previous trajectory matching works (Du et al., 2022; Guo et al., 2024; Li et al., 2024) to train expert trajectories for one hundred epochs. Hyper-parameters are directly adopted without

---

**Algorithm 1** Emphasizing Discriminative Features

---

1: **Input:** $D_{real}$: The real dataset
2: **Input:** $D_{syn}$: The synthetic dataset
3: **Input:** $\mathcal{A}$: A trajectory-matching based algorithm
4: **Input:** $\mathcal{G}$: Grad-CAM model
5: **Input:** $K$: Activation maps update frequency
6: **Input:** $\alpha$: Threshold of supervision dropout
7: **Input:** $T$: Total distillation steps
8: **Input:** $\beta$: Enhancement factor
9: **Input:** $\mathcal{F}$: Activation map processing function
10: **Input:** $r$: Learning rate of synthetic dataset
11: **for** $t$ in $0 \ldots T-1$ **do**
12:     $L \leftarrow \mathcal{A}(D_{syn}, D_{real})$      ▷ Compute the array of trajectory matching losses
13:     $L' \leftarrow Sort(L)$      ▷ Sort $L$ to get ordered losses
14:     $L_{edf} \leftarrow \sum_{i=\alpha|L|}^{|L|} L'_i$      ▷ Dropout low-loss supervision
15:     **if** $t \bmod K = 0$ **then**
16:        $M \leftarrow \mathcal{G}(D_{syn})$      ▷ Update activation maps of current $S$
17:     **end if**
18:     $(\nabla D_{syn})_{EDF} \leftarrow \nabla D_{syn} \circ \mathcal{F}(M, \beta)$      ▷ Process synthetic image gradients
19:     $D_{syn} \leftarrow D_{syn} - r \cdot (\nabla D_{syn})_{EDF}$      ▷ Biased update towards discriminative areas
20: **end for**
21: Return $D_{syn}$

---

modification. For distillation, we implement EDF based on DATM (Guo et al., 2024) and PAD (Li et al., 2024), which simultaneously distills soft labels along with images.

We use torch-cam (Fernandez, 2020) for Grad-CAM implementation. Hyper-parameters are listed in Table 8.

## B.2 EVALUATION DETAILS

To achieve a fair comparison, when comparing EDF with DD methods, we only adopt the set of differentiable augmentations commonly used in previous studies (Zhao & Bilen, 2021b;a; Cazenavette et al., 2022) to train a surrogate model on distilled data and labels.

When comparing EDF with DD+KD methods, we follow their evaluation methods, which we detail the steps as follows:

1. Train a teacher model on the real dataset and freeze it afterward.

2. Train a student model on the distilled dataset by **minimizing the KL-Divergence loss** between the output of the student model and the output of the teacher model on the same batch from distilled data.

3. Validate the student model on the test set and obtain test accuracy.

For implementation, please refer to the official repo of SRe2L[1] and RDED[2].

## B.3 COMPUTING RESOURCES

Experiments on IPC 1/10 can be run with 4x Nvidia-A100 80GB GPUs, and experiments on IPC 50 can be run with 8x Nvidia-A100 80GB GPUs. The GPU memory demand is primarily dictated by the volume of synthetic data per batch and the total training iterations the augmentation model undergoes with that data. When IPC becomes large, GPU usage can be optimized by either adopting techniques like TESLA (Cui et al., 2022a) or by scaling down the number of training iterations ("syn_steps") or shrinking the synthetic data batch size ("batch_syn").

---

[1] https://github.com/VILA-Lab/SRe2L/tree/main/SRe2L
[2] https://github.com/LINs-lab/RDED

| Modules Hyper-parameters | CPD $\alpha$ | DAE $\beta$ | $K$ | $T$ | TM batch_syn | lr_pixel | lr_label | syn_steps |
|---|---|---|---|---|---|---|---|---|
| | 1 | 0 | 1 | 50 | | 1000 | 10000 | 2.0 | |
| ImageNette 10 | 0.25 | 1 | 100 | 10000 | 400 | 1000 | 2.0 | 40 |
| 50 | 0.375 | 2 | 200 | | 200 | 100 | 5.0 | |
| | 1 | 0 | 1 | 50 | | 1000 | 10000 | 2.0 | |
| ImageWoof 10 | 0.25 | 2 | 100 | 10000 | 400 | 1000 | 2.0 | 40 |
| 50 | 0.375 | 1 | 200 | | 200 | 100 | 5.0 | |
| | 1 | 0 | 1 | 50 | | 1000 | 10000 | 3.0 | |
| ImageMeow 10 | 0.25 | 1 | 100 | 10000 | 400 | 1000 | 2.0 | 40 |
| 50 | 0.375 | 2 | 200 | | 200 | 100 | 5.0 | |
| | 1 | 0 | 1 | 50 | | 1000 | 10000 | 3.0 | |
| ImageYellow 10 | 0.25 | 1 | 100 | 10000 | 400 | 1000 | 3.0 | 40 |
| 50 | 0.375 | 2 | 200 | | 200 | 100 | 5.0 | |
| | 1 | 0 | 1 | 50 | | 1000 | 10000 | 3.0 | |
| ImageFruit 10 | 0.25 | 1 | 100 | 10000 | 400 | 1000 | 2.0 | 40 |
| 50 | 0.375 | 2 | 200 | | 200 | 100 | 5.0 | |
| | 1 | 0 | 1 | 50 | | 1000 | 10000 | 3.0 | |
| ImageSquawk 10 | 0.25 | 1 | 100 | 10000 | 400 | 1000 | 3.0 | 40 |
| 50 | 0.375 | 2 | 200 | | 200 | 100 | 5.0 | |

Table 8: Hyper-parameters of experiments on ImageNet-1K and nette, woof, meow, fruit, yellow, squawk subsets.

| Method | ConvNetD5 | ResNet18 | VGG11 | AlexNet |
|---|---|---|---|---|
| Random | 41.8 | 40.9 | 43.2 | 35.7 |
| FTD | 62.8 | 49.8 | 50.5 | 47.6 |
| DATM | 65.1 | **52.4** | 51.2 | **49.6** |
| **EDF** | **68.2** | 50.8 | **53.2** | 48.2 |

(a) ImageYellow, IPC10

| Method | ConvNetD5 | ResNet18 | VGG11 | AlexNet |
|---|---|---|---|---|
| Random | 29.6 | 31.4 | 30.8 | 25.7 |
| FTD | 58.4 | 55.6 | 57.6 | 52.3 |
| DATM | 61.8 | 62.8 | **65.6** | 63.5 |
| **EDF** | **65.4** | **63.6** | 64.8 | **69.2** |

(b) ImageSquawk, IPC50

Table 9: Cross-architecture evaluation on ResNet18, VGG11, and AlexNet. ConvNetD5 is the distillation architecture. Distilled datasets under IPC10 and IPC50 outperform random selection, FTD, and DATM, showing good generalizability.

## C  ADDITIONAL EXPERIMENT RESULTS AND FINDINGS

### C.1  CROSS-ARCHITECTURE EVALUATION

Generalizability on different model architectures is one key property of a well-distilled dataset. To show that EDF can generalize well on different models, we evaluate synthetic images under IPC 10 and 50 of the ImageSquawk subset, on three other standard models, AlexNet (Krizhevsky et al., 2012), VGG11 (Simonyan & Zisserman, 2014), and ResNet18 (He et al., 2015). As shown in Table 11, our distilled datasets outperform random selection and two baseline methods on both IPC10 and IPC50. Compared with IPC10, distilled images under IPC50 can achieve better performance on unseen neural networks. This suggests that EDF's distillation results have decent generalizability across different architectures, especially when the compressing ratio is smaller which allows distilled datasets to accommodate more discriminative information.

### C.2  EVAL. WITHOUT KNOWLEDGE DISTILLATION

Starting from Wang et al. (2020), representative dataset distillation (DD) methods (Zhao et al., 2021; Zhao & Bilen, 2021b; Cazenavette et al., 2022; Wang et al., 2022) establish a general workflow as follows: 1) *Distillation*: At this stage, information from the real dataset is fully accessible to the DD algorithm to train synthetic data. 2) *Evaluation*: After the distilled dataset is obtained, the evaluation

| Dataset | ImageNette | | | ImageWoof | | | ImageSquawk | | |
|---|---|---|---|---|---|---|---|---|---|
| IPC | 1 | 10 | 50 | 1 | 10 | 50 | 1 | 10 | 50 |
| SRe2L | 18.4±0.8 | 41.0±0.3 | 55.6±0.2 | 16.0±0.2 | 32.2±0.3 | 35.8±0.2 | 22.5±0.5 | 35.6±0.4 | 42.2±0.3 |
| RDED | 28.0±0.5 | 53.6±0.8 | 72.8±0.3 | 19.0±0.3 | 32.6±0.5 | **52.6±0.6** | 33.8±0.5 | 52.2±0.5 | 71.6±0.8 |
| EDF | **52.6±0.5** | **71.0±0.8** | **77.8±0.5** | **30.8±1.0** | **41.8±0.2** | 48.4±0.5 | **41.8±0.5** | **65.4±0.8** | **74.8±1.2** |

Table 10: Performances of SRe2L and RDED without using knowledge distillation during evaluation. EDF outperforms the other two methods in most of settings, and our advantage is more pronounced as IPC gets smaller.

is performed by training a randomly initialized model on the distilled data. Specifically, in the context of classification, the objective is to minimize cross-entropy loss. Recently, some new methods (Yin et al., 2023; Sun et al., 2023) introduced teacher knowledge into the student model by applying knowledge distillation. Although it helps improve performances to a large extent, it may not be able to reflect the effectiveness of dataset distillation accurately.

To this end, we remove the knowledge distillation from Eval. w/ Knowledge Distillation (SRe2L and RDED) methods but keep soft labels to ensure a fair comparison, Specifically, we train a classification model on the synthetic images by only minimizing the cross-entropy loss between student output and soft labels. As shown in Table 10, without knowledge distillation, EDF outperforms SRe2L and RDED in 8 out of 9 settings. Our advantage is more pronounced, especially when IPC is smaller, underscoring the superior efficacy of EDF on smaller compressing ratios.

### C.3 DISTORTED IMAGES OF LARGE ENHANCEMENT FACTOR

In Figure 6, we show results of using excessively large enhancement factors as mentioned in Section 4.3. The distributions of these distilled images are distorted, with many pixels containing only blurred information. This occurs because excessively increasing the gradients in discriminative areas can lead to large updates between iterations, resulting in the divergence of the pixel distribution. Therefore, the enhancement of discrimination areas is not the stronger the better. It is important to maintain the enhancement factor within a reasonable range.

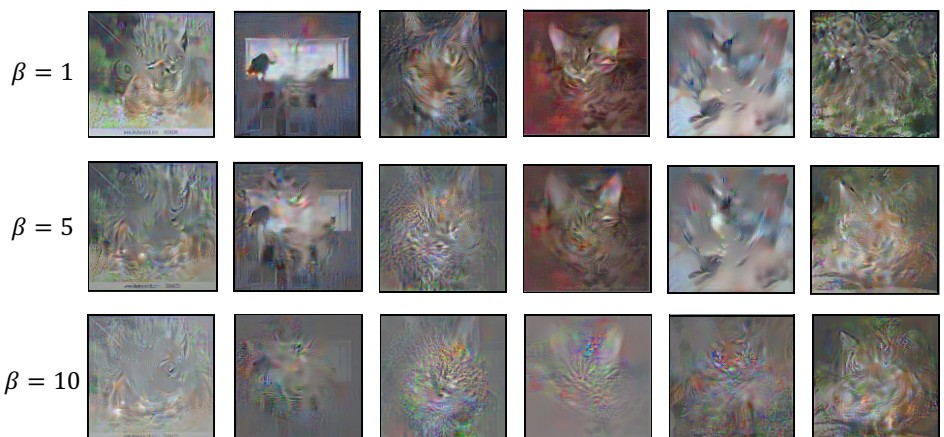

Figure 6: Distorted image distributions due to excessively large enhancement factors (= 10)

## D COMP-DD BENCHMARK

### D.1 SUBSET DETAILS

The corresponding class labels for each subset are listed as follows:

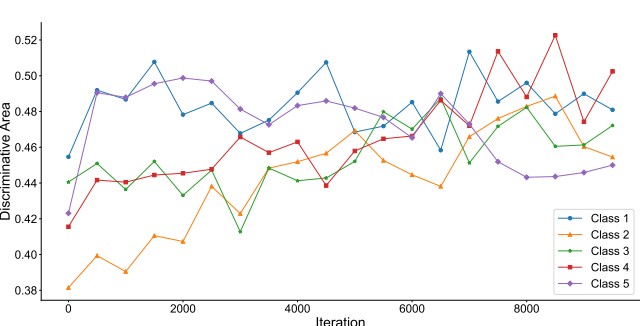 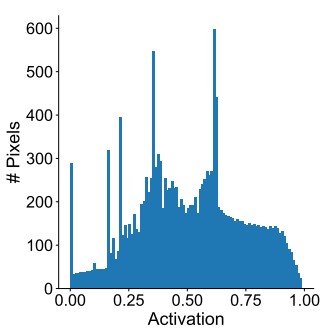

(a) In general, discriminative areas show a trend of increase as the distillation proceeds.

(b) Most of the pixels have activation around 0.25 to 0.75.

Figure 7: **(a)** The trend of discriminative area change across various distillation iterations. **(b)** Distribution of the activation map of a random image from ImageNet-1K.

- **Bird-Hard:** n01537544, n01592084, n01824575, n01558993, n01534433, n01843065, n01530575, n01560419, n01601694, n01532829
- **Bird-Easy:** n02007558, n02027492, n01798484, n02033041, n02012849, n02025239, n01818515, n01820546, n02051845, n01608432
- **Dog-Hard:** n02107683, n02107574, n02109525, n02096585, n02085620, n02113712, n02086910, n02093647, n02086079, n02102040
- **Dog-Easy:** n02096294, n02093428, n02105412, n02089973, n02109047, n02109961, n02105056, n02092002, n02114367, n02110627
- **Car-Hard:** n04252077, n03776460, n04335435, n03670208, n03594945, n03445924, n03444034, n04467665, n03977966, n02704792
- **Car-Easy:** n03459775, n03208938, n03930630, n04285008, n03100240, n02814533, n03770679, n04065272, n03777568, n04037443
- **Snake-Hard:** n01693334, n01687978, n01685808, n01682714, n01688243, n01737021, n01751748, n01739381, n01728920, n01728572
- **Snake-Easy:** n01749939, n01735189, n01729977, n01734418, n01742172, n01744401, n01756291, n01755581, n01729322, n01740131
- **Insect-Hard:** n02165456, n02281787, n02280649, n02172182, n02281406, n02165105, n02264363, n02268853, n01770081, n02277742
- **Insect-Easy:** n02279972, n02233338, n02219486, n02206856, n02174001, n02190166, n02167151, n02231487, n02168699, n02236044
- **Fish-Hard:** n01440764, n02536864, n02514041, n02641379, n01494475, n02643566, n01484850, n02640242, n01698640, n01873310
- **Fish-Easy:** n01496331, n01443537, n01498041, n02655020, n02526121, n01491361, n02606052, n02607072, n02071294, n02066245
- **Round-Hard:** n04409515, n04254680, n03982430, n04548280, n02799071, n03445777, n03942813, n03134739, n04039381, n09229709
- **Round-Easy:** n02782093, n03379051, n07753275, n04328186, n02794156, n09835506, n02802426, n04540053, n04019541, n04118538
- **Music-Hard:** n02787622, n03495258, n02787622, n03452741, n02676566, n04141076, n02992211, n02672831, n03272010, n03372029
- **Music-Easy:** n03250847, n03854065, n03017168, n03394916, n03721384, n03110669, n04487394, n03838899, n04536866, n04515003

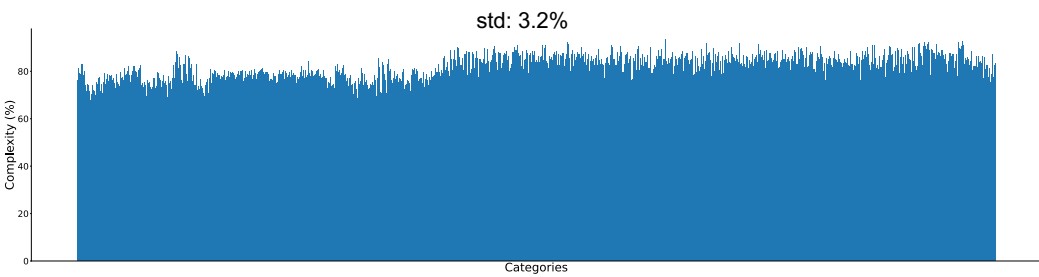

Figure 8: Complexity distribution of all classes from ImageNet-1K under threshold being 0.1.

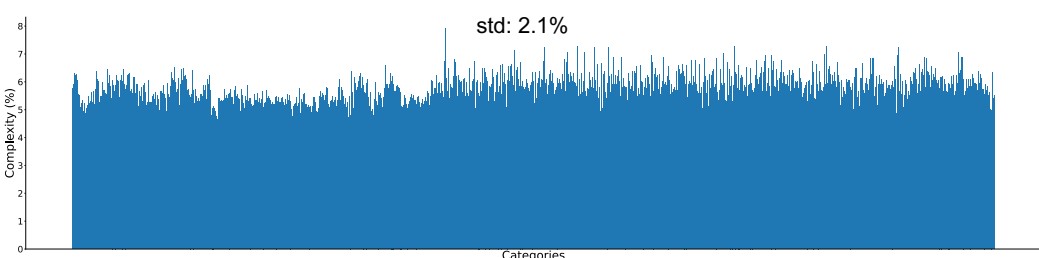

Figure 9: Complexity distribution of all classes from ImageNet-1K under threshold being 0.9.

## D.2 COMPLEXITY METRICS

We use the percentage of pixels whose Grad-CAM activation values exceed a predefined fixed threshold to evaluate the complexity of an image. In our settings, the fixed threshold is 0.5. The reasons for fixing the threshold at 0.5 are twofold. Firstly, when selecting subsets, images are static and won't be updated in any form (this is different from EDF's DAE module, which updates synthetic images). Thus, using a fixed threshold is sufficient for determining the high-activation areas.

Secondly, values of a Grad-CAM activation map range from 0 to 1, with higher values corresponding to higher activation. We present the distribution of the activation map of a random image from ImageNet-1K in Figure 13b, where the majority of pixels have activation values between 0.25 and 0.75. Subsequently, if the threshold is too small or too large, the complexity scores of all classes will be close (standard deviation is small), as shown in Figure 12 and 13. This results in no clear distinguishment between easy and hard subsets. Finally, we set 0.5 as the threshold, which is the middle point of the range. Complexity distribution under this threshold is shown in Figure 10.

Our complexity metrics are an early effort to define how complex an image is in the context of dataset distillation. We acknowledge potential biases or disadvantages and encourage future studies to continue the refinement of complex metrics.

## D.3 BENCHMARK HYPER-PARAMETERS

For the trajectory training, experiment settings are the same as those used for ImageNet-1K and its subsets. For distillation, we provide hyper-parameters of EDF on the Complex DD Benchmark in Table 11. These hyper-parameters can serve as a reference for future works to extend to other subsets of the benchmark.

## E VISUALIZATION OF DISTILLED IMAGES ON IMAGENET-1K

In Figure 11 to 13, we present a visualization of distilled images of all ImageNet-1K subsets in Table 1.

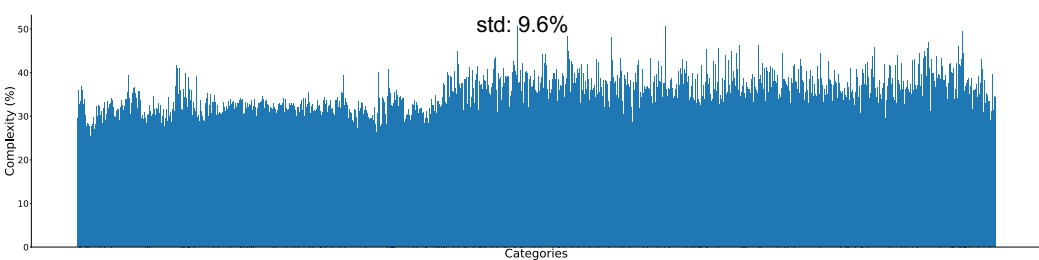

Figure 10: Complexity distribution of all classes from ImageNet-1K under threshold being 0.5.

| Modules | | CPD | DAE | | | TM | | | | |
|---|---|---|---|---|---|---|---|---|---|---|
| Hyper-parameters | | $\alpha$ | $\beta$ | $K$ | $T$ | batch_syn | lr_pixel | lr_label | syn_steps |
| CDD-Bird-Easy | 1 | 0 | 1 | 50 | | 1000 | 10000 | 2.0 | |
| | 10 | 0.25 | 1 | 100 | 10000 | 400 | 1000 | 3.0 | 40 |
| | 50 | 0.375 | 2 | 200 | | 200 | 100 | 5.0 | |
| CDD-Bird-Hard | 1 | 0 | 1 | 50 | | 1000 | 10000 | 2.0 | |
| | 10 | 0.25 | 1 | 100 | 10000 | 400 | 1000 | 3.0 | 40 |
| | 50 | 0.375 | 2 | 200 | | 200 | 100 | 5.0 | |
| CDD-Dog-Easy | 1 | 0 | 1 | 50 | | 1000 | 10000 | 2.0 | |
| | 10 | 0.25 | 1 | 100 | 10000 | 400 | 1000 | 5.0 | 40 |
| | 50 | 0.375 | 2 | 200 | | 200 | 100 | 5.0 | |
| CDD-Dog-Hard | 1 | 0 | 1 | 50 | | 1000 | 10000 | 2.0 | |
| | 10 | 0.25 | 1 | 100 | 10000 | 400 | 1000 | 2.0 | 40 |
| | 50 | 0.375 | 2 | 200 | | 200 | 100 | 5.0 | |
| CDD-Car-Easy | 1 | 0 | 1 | 50 | | 1000 | 10000 | 3.0 | |
| | 10 | 0.25 | 1 | 100 | 10000 | 400 | 1000 | 3.0 | 40 |
| | 50 | 0.375 | 2 | 200 | | 200 | 100 | 5.0 | |
| CDD-Car-Hard | 1 | 0 | 1 | 50 | | 1000 | 10000 | 3.0 | |
| | 10 | 0.25 | 1 | 100 | 10000 | 400 | 1000 | 3.0 | 40 |
| | 50 | 0.375 | 2 | 200 | | 200 | 100 | 5.0 | |

Table 11: Hyper-parameters of EDF on the Complex DD Benchmark.

# F    MORE RELATED WORK

In Table 12, we present a comprehensive summary of previous dataset distillation methods, categorized by different approaches. There are four main categories of dataset distillation: gradient matching, trajectory matching, distribution matching, and generative model-based methods. Recently, some works (Yin et al., 2023; Sun et al., 2023; Yu et al., 2024) add knowledge distillation during the evaluation stage of dataset distillation.

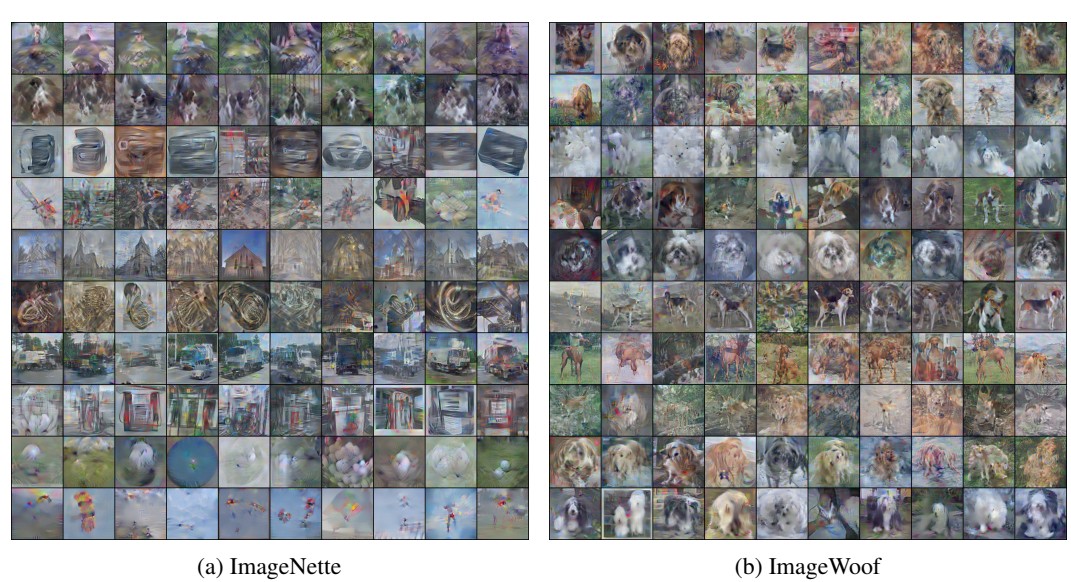

(a) ImageNette

(b) ImageWoof

Figure 11

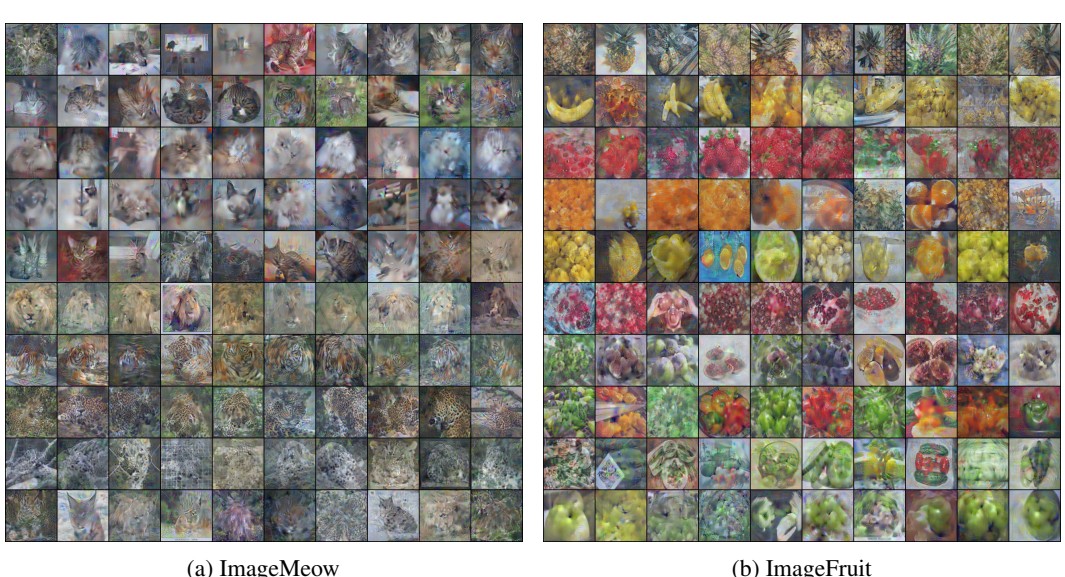

(a) ImageMeow

(b) ImageFruit

Figure 12

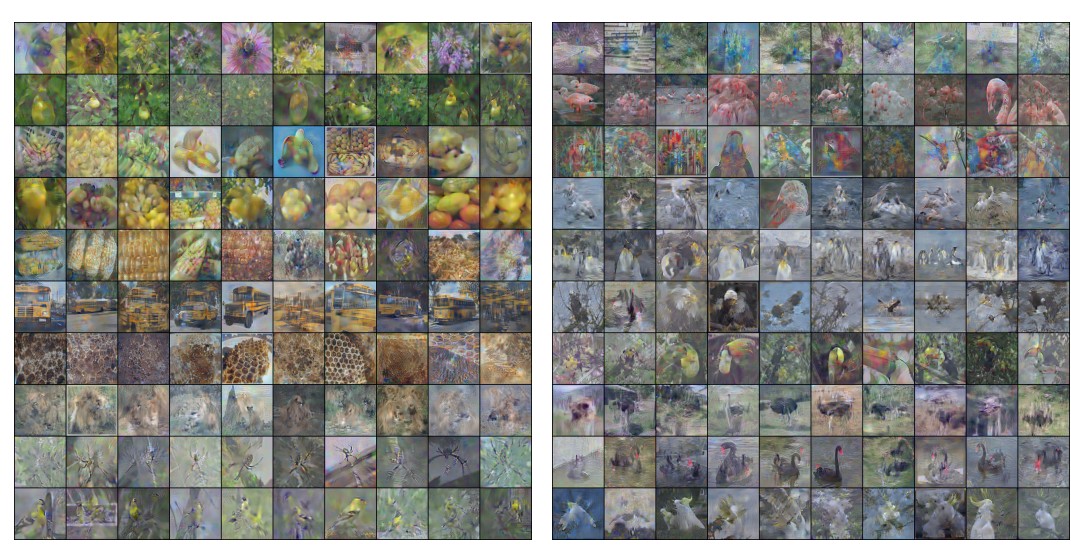

(a) ImageYellow                                    (b) ImageSquawk

Figure 13

| Category | Method |
|---|---|
| Gradient-matching | DC (Zhao et al., 2021)
DSA (Zhao & Bilen, 2021b)
DCC (Lee et al., 2022a)
LCMat (Shin et al., 2023) |
| Trajectory-matching | MTT (Cazenavette et al., 2022)
Tesla (Cui et al., 2022a)
FTD (Du et al., 2022)
SeqMatch (Du et al., 2023)
DATM (Guo et al., 2024)
ATT (Liu et al., 2024)
NSD (Yang et al., 2024)
PAD (Li et al., 2024)
SelMatch (Lee & Chung, 2024) |
| Distribution-matching | DM (Zhao & Bilen, 2021a)
CAFE (Wang et al., 2022)
IDM (Zhao et al., 2023)
DREAM (Liu et al., 2023b)
M3D (Zhang et al., 2023) |
| Generative model | DiM Wang et al. (2023)
GLaD (Cazenavette et al., 2023)
H-GLaD (Zhong et al., 2024)
LD3M (Moser et al., 2024)
IT-GAN (Zhao & Bilen, 2022)
D4M Su et al. (2024)
Minimax Diffusion Gu et al. (2023) |
| + Knowledge distillation for evaluation | SRe2L (Yin et al., 2023)
RDED (Sun et al., 2023)
HeLIO (Yu et al., 2024) |

Table 12: Summary of previous works on dataset distillation

