# OpenReview forum: "Emphasizing Discriminative Features for Dataset Distillation in Complex Scenarios"
_ICLR.cc/2025/Conference — ICLR 2025 Conference Withdrawn Submission_

### Official Review · Reviewer_MCxt · 2024-10-30

**Soundness:** 3
**Presentation:** 3
**Contribution:** 3
**Rating:** 5
**Confidence:** 3

**Summary:**

This manuscript introduces a data distillation approach, called EDF, by emphasizing the discriminative features using Grad-CAM. Through the various experimental observations, the EDF enhances the high activated regions. Experimental results show the effectiveness of the proposed method.

**Strengths:**

__S1.__ Through experimental observations (comparison between low-loss and high-loss, t-SNE, Grad-CAM) in the introduction, the authors' introduction of EDF is convincing to the readers.

__S2.__ Overall manuscript is easy to understand to me except some concerns raised by the weakness.

__S3.__ To validate the proposed method the authors introduce a complex DD benckmark.

**Weaknesses:**

__W1.__ What is the exact meaning of 'common pattern'? More detailed explanation is needed.

__W2.__ The statement "low-loss signals typically correspond to common patterns" lacks specific reasoning to explain why dropout should be applied to address this. Can the authors elaborate on the rationale for using dropout?

__W3.__ Observations were made in the introduction; however, they were based on CIFAR10 and IN1K-CIFAR10. Do similar results generally appear on other datasets as well, without cherry-picking?

__W4.__ In a complex scene, object sizes are generally small, which could potentially be a reason for the smaller regions.

__W5.__ Can the proposed method be applied to other methods besides Grad-CAM?

__W6.__ How can the authors be certain that this is a discriminative area? This part is difficult to understand, and an explanation and justification are needed.

__W7.__ What happens in a general environment (like the one conducted in DATM) rather than a complex DD benchmark?

**Questions:**

Specific reasoning is needed for some words and justifications. Additionally, evidence is required to support that the claims are not dataset-dependent. The comment mentioned in the weaknesses is my main concern.

---

### Official Review · Reviewer_29Q1 · 2024-11-02

**Soundness:** 2
**Presentation:** 2
**Contribution:** 3
**Rating:** 6
**Confidence:** 4

**Summary:**

This paper presents a dataset distillation method called Emphasize Discriminative Features (EDF), which aims to enhance the effectiveness of synthetic datasets in complex situations. EDF contains two components, common pattern dropout (CPD) and discriminative area enhancement (DAE). CPD selectively keeps the parameters with greater difference during the trajectory matching, while DAE targets discriminative regions within images by leveraging class activation maps. Additionally, the authors propose a new benchmark, Comp-DD, aimed at evaluating dataset distillation performance with varying complexity levels.

**Strengths:**

This work addresses a key challenge of applying dataset distillation to complex image datasets. This focus on complex scenarios has the potential to significantly advance the practicality and real-world applicability of DD methods.

**Weaknesses:**

- Some details of the proposed method are missing, as noted in the questions section below.
- This work introduces many hyperparameters, i.e. target supervision dropout ratio ($\alpha$), activation map update frequencies ($K$), and enhancement factor ($\beta$). A more in-depth analysis of the hyperparameter space and its influence on performance would improve the method's usability and adaptability.

**Questions:**

- In Equation (1), do low-loss elements represent common features? There are two possibilities—either a small numerator or a large denominator—which seem to imply different meanings.
- How does the proposed EDF method perform on simpler datasets, such as CIFAR-10 or CIFAR-100, compared to established dataset distillation baselines like MTT, FTD, and DATM?
- The paper mentions the use of "Eval. w/ Knowledge Distillation" as a new evaluation component. However, it lacks a detailed explanation of this component. Could the authors elaborate on evaluation approach?
- In Algorithm 1, what is the specific relationship between the dropout of low-loss supervision ($L_{edf}$) and the calculation of synthetic image gradients ($\nabla D_{syn}$)?
- In line 170, the authors sort the parameters based on their loss value, and the smallest $\alpha \times P$ parameters are dropped out. However, EDF method in Table 7 shows that the parameters in layers 2, 4, 5, and 6 were dropped. Could the authors clarify this discrepancy?

---

### Official Review · Reviewer_GwUt · 2024-11-03

**Soundness:** 2
**Presentation:** 3
**Contribution:** 2
**Rating:** 5
**Confidence:** 4

**Summary:**

### Summary

This paper presents a new method (EDF) for dataset distillation targeting improved performance in complex scenarios like ImageNet-1K subsets. EDF integrates Grad-CAM activation maps to focus on high-activation, discriminative regions within synthetic data, unlike previous methods that treat all image pixels equally. It also introduces the *Common Pattern Dropout* module to remove low-loss supervision signals, which often contain non-discriminative common patterns. Additionally, the authors contribute a new benchmark, *Complex Dataset Distillation (Comp-DD)*, to help the community evaluate distillation methods in complex scenarios. Experimental results show EDF outperforming state-of-the-art (SOTA) methods in challenging datasets and achieving "lossless" performance on some subsets.

**Strengths:**

- **Novelty**: EDF’s use of Grad-CAM to target high-activation areas in synthetic images is innovative, particularly for complex datasets where discriminative features are limited to small regions. The *Common Pattern Dropout* module’s filtering of low-loss signals is a creative solution in an attempt to minimize non-discriminative features, improving the overall quality of synthetic data.

- **Benchmark Contribution**: The creation of the Comp-DD benchmark provides a valuable tool for future work, establishing a way to test dataset distillation methods on complex scenarios.

- **Strong Results**: EDF achieves SOTA performance across a variety of complex dataset subsets, with substantial gains in accuracy and the ability to achieve near-lossless performance in certain settings.

**Weaknesses:**

- **Inaccuracy of Grad-CAM Definition in Section 2.2**: According to the original Grad-CAM paper, $M^c$ should be a gradient-weighted sum of all feature maps in the last convolutional layer. Therefore, the symbol $l$ in Equation (3) should denote the $l$-th feature map instead of the $l$-th convolutional layer.

**Selvaraju, R.R., Cogswell, M., Das, A., Vedantam, R., Parikh, D., & Batra, D.** (2017). *Grad-CAM: Visual Explanations from Deep Networks via Gradient-Based Localization.* In *Proceedings of the IEEE International Conference on Computer Vision* (pp. 618-626). doi:[10.1109/ICCV.2017.74](https://doi.org/10.1109/ICCV.2017.74)

- **Dependency on Hyperparameters**: EDF’s performance depends heavily on various hyperparameters (dropout ratio $\alpha$, Grad-CAM update frequency $K$, enhancement factor $\beta$), requiring fine-tuning for a particular dataset, which could limit usability in practical applications without extensive experimentation.

- **Generalizability to Non-Complex Scenarios**: EDF shows significant improvements on complex datasets; however, its benefits on simpler datasets (like CIFAR) are less clear. It’s valuable to show EDF’s generalizability across datasets of various complexity.

- **Limited Complexity of Comp-DD**: ImageNet is a curated dataset, with most object-related images in a portrait style. Consequently, the subsets chosen from ImageNet in Comp-DD are likely also portrait-oriented with larger background regions. This setup may limit the true complexity of the benchmark. To further increase complexity, incorporating zoomed-out, cropped objects with relevant labels from uncurated datasets such as COCO, Objects365, or SA-1B could provide a more challenging and diverse set of scenarios, better suited to the intended purpose.

- **Model Bias in Generating Activation Maps**: It is well-known that neural networks can capture biased or non-generalizable features, as discussed in the Grad-CAM paper. If the activation maps focus on such less-generalizable regions, this could undermine the effectiveness of the *Discriminative Area Enhancement (DAE)* module. However, it is unclear how the proposed method addresses this issue of potential bias in the activation maps. Clarification on how EDF mitigates or adapts to these biased features would strengthen the approach.

- **Clarification on Low-Loss Regions and Discriminative Features**: The paper assumes that low-loss regions correspond to common, less-discriminative features. Although the paper demonstrates activation area shifts with different loss levels during distillation (Fig. 2a), additional qualitative results—such as visual analysis with superposed images and highlighted activation regions—would help support this assumption. It is important to note that a smaller activation region does not necessarily indicate less-discriminative features.

Recommendation:
Overall, my recommendation for this paper is borderline reject. However, the score could be improved if the concerns outlined above are properly addressed.

**Questions:**

None

---

### Official Review · Reviewer_AX7Z · 2024-11-05

**Soundness:** 2
**Presentation:** 3
**Contribution:** 1
**Rating:** 3
**Confidence:** 4

**Summary:**

This paper introduces a dataset distillation method termed EDF (emphasizes the discriminative features). The authors leverage Grad-CAM activation maps to enhance key discriminative regions and downplay low-loss supervision signals to avoid common patterns in synthetic images. This work also constructs the Complex Dataset Distillation (Comp-DD) benchmark from ImageNet-1K to facilitate research in complex scenarios. Extensive experiments demonstrate the effectiveness of the proposed EDF.

**Strengths:**

1. The motivation is well-understood. The EDF method sounds reasonable.
2. The evaluation is extensive. The experimental settings are convincing. The analysis is thorough.
3. The proposed Comp-DD benchmark contributes to community research.
4. The paper is well-organized and easy to follow.

**Weaknesses:**

1. Using the original image for initialization seems to be critical. As indicated by Figure 5, discriminative features are highly contingent on the initialized image. If random initialization is employed, the calculation of Grad-CAM
may lose its significance and potentially lead to convergence issues, whereas the baseline method MTT [1] is viable under random noise initialization. The authors should include experiments involving random noise initialization.
2. This method seems to be a combination of DATM and Grad-CAM. While the authors demonstrate the advantages of Grad-CAM over DATM [2], the versatility of this method requires further validation. Can EDF be integrated with additional
baseline methods such as DC [3], DM [4], CAFE [5], and DREAM [6]?
3. The authors should explicitly provide the proportion of wall clock time allocated for calculating Grad-CAM during training. Furthermore, can EDF surpass that of DATM when both methods are trained within the same wall clock time?
4. The network utilized for Grad-CAM calculation must first be trained on the same real dataset. Thus, its inherent bias may act as a bottleneck for gradient updates. The authors can conduct an ablation study to evaluate the impact of
 the network's capabilities on EDF.
5. It is worth discussing how the authors compare EDF’s approach, which focuses on highly activated regions, with previous methods aimed at reducing spatial redundancy [7][8] and learning a highly representative basis [9].
6. As shown in Table 11, each IPC configuration corresponds to distinct hyperparameters, which may complicate practical applications, especially given that MTT [1] and DATM [2] already involve numerous hyperparameters that require
tuning.
7. The authors claim to be the first to achieve lossless results on ImageNet subsets; however, based on the trends depicted in Table 1 and the characteristics of DATM [2], it can be inferred that DATM may also attain lossless results
at IPC 200/300. It becomes necessary to supplement experimental validation and accuracy comparisons.
8. I'm curious whether cross-architecture evaluations can achieve lossless results, such as with ResNet50 and other advanced networks. If lossless results are attained only on less robust networks (e.g., ConvNetD5), the practical
significance may be limited due to lower performance ceilings. Recent studies have reported higher test accuracy on larger networks [10][11][12][13].

[1] "Dataset distillation by matching training trajectories." CVPR 2022.

[2] "Towards Lossless Dataset Distillation via Difficulty-Aligned Trajectory Matching." ICLR 2024.

[3] "Dataset condensation with gradient matching." ICLR 2021.

[4] "Dataset Condensation with Distribution Matching." arXiv preprint arXiv:2110.04181 (2021).

[5] "Cafe: Learning to condense dataset by aligning features." CVPR 2022.

[6] "Dream: Efficient dataset distillation by representative matching." ICCV 2023.

[7] "Dataset condensation via efficient synthetic-data parameterization." ICML 2022.

[8] "Sparse parameterization for epitomic dataset distillation." NeurIPS 2023.

[9] "Dataset distillation via factorization." NeurIPS 2022.

[10] "Squeeze, recover and relabel: Dataset condensation at imagenet scale from a new perspective." NeurIPS 2023.

[11] "Dataset Distillation in Large Data Era." arXiv preprint arXiv:2311.18838 (2023).

[12] "Curriculum Dataset Distillation." arXiv preprint arXiv:2405.09150 (2024).

[13] "Large Scale Dataset Distillation with Domain Shift." ICML 2024.

**Questions:**

Please see the weaknesses above.

---

### Note · Authors · 2024-11-13

I have read and agree with the venue's withdrawal policy on behalf of myself and my co-authors.